# PLGA-Based Composites for Various Biomedical Applications

**DOI:** 10.3390/ijms23042034

**Published:** 2022-02-12

**Authors:** Cátia Vieira Rocha, Victor Gonçalves, Milene Costa da Silva, Manuel Bañobre-López, Juan Gallo

**Affiliations:** Advanced (Magnetic) Theranostic Nanostructures Lab, Health Cluster, International Iberian Nanotechnology Laboratory, Av. Mestre José Veiga s/n, 4715-330 Braga, Portugal; catia.rocha@inl.int (C.V.R.); victor.goncalves@inl.int (V.G.); milene.silva@inl.int (M.C.d.S.); manuel.banobre@inl.int (M.B.-L.)

**Keywords:** PLGA, composites, inorganic nanoparticles, scaffolds, biomedical applications

## Abstract

Polymeric materials have been extensively explored in the field of nanomedicine; within them, poly lactic-co-glycolic acid (PLGA) holds a prominent position in micro- and nanotechnology due to its biocompatibility and controllable biodegradability. In this review we focus on the combination of PLGA with different inorganic nanomaterials in the form of nanocomposites to overcome the polymer’s limitations and extend its field of applications. We discuss their physicochemical properties and a variety of well-established synthesis methods for the preparation of different PLGA-based materials. Recent progress in the design and biomedical applications of PLGA-based materials are thoroughly discussed to provide a framework for future research.

## 1. Introduction

Polymeric materials have become extremely attractive materials for a wide range of applications. One of the areas where they are becoming increasingly important is biomedicine [1]. One extensively explored polymer is poly lactic-co-glycolic acid (PLGA), a biodegradable, biocompatible and FDA (U.S. Food and Drug Administration, USA)/EMA (European Medicines Agency)-approved copolymer [2,3]. It is commercially available with different molecular weights and copolymer ratios, which allow for the tuning of the final PLGA behavior to fit an application of interest [4,5]. Being the scope of this review of biomedical applications, PLGA features that can be useful in this area will be explored the most. PLGA can be processed into any shape and size, has great water solubility and allows for a tunable drug release. Its biodistribution and pharmacokinetics follow nonlinear and dose-dependent profiles [2,6]. Generally speaking, copolymers can be synthesized as random or block copolymers, presenting different intrinsic properties in this way. PLGA is generally synthesized through the ring-opening copolymerization of lactic acid (LA) and glycolic acid (GA), with the products of its degradation being nontoxic [7,8,9]. Normally, following the aforementioned synthesis technique, randomly distributed PLGA can be obtained in either: (i) an atactic configuration where the repeating units have no regular stereochemical configuration or (ii) a syndiotactic configuration where the repeating units have alternating stereochemical configurations [10]. Sequencing is important since it considerably influences the degradation rate of PLGA, with random PLGA degrading much quicker than sequenced PLGA [11].

Over the last decade PLGA copolymers have been extensively explored in combination with nanotechnology. Nanotechnology brings unique characteristics to the system, which, in combination with PLGA’s unique features, allow for extensive biomedical applications. It is possible to encapsulate both organic and inorganic materials into PLGA, such as small-molecule drugs [2,12], vaccines [13], proteins [14,15] and metallic [16] as well as magnetic [17,18] nanoparticles (NPs). The methods of production of PLGA-based materials can be adapted to various types of drugs, making it possible to encapsulate hydrophobic and hydrophilic molecules and thus making this copolymer an ideal drug delivery system (DDS) [4].

PLGA-based technology has been used for many years for a myriad of applications in the biomedical arena, with many being already approved by the FDA and EMA (Table 1). Drug delivery is one of the main uses for PLGA, which can be accomplished in the form of macroscopic structures (scaffolds/gels), microparticles (MPs) or NPs. In general, in the biomedical field these PLGA structures have proved their potential for encapsulating several therapeutic agents towards different ends (e.g., antiseptics [19], antibiotics [20], anti-inflammatory [21] and antioxidant [22] drugs), and have shown promise for specific targeting when adequately functionalized [4,23]. On top of that, PLGA-based materials protect their cargo from degradation and provide a sustained drug release profile, ideal for long-term treatments [4,7].

The combination of PLGA with inorganic NPs (INPs) to form nanocomposite materials has the potential to extend the field of the application of this polymer even further and potentiate the effect of these materials while maintaining a reduced toxicity. There is a wide range of INPs available to be combined with PLGA, and by doing so some of the limitations of INPs, such as their toxicity and colloidal stability, can be circumvented. For these reasons the applications of PLGA-based nanocomposite materials in chemotherapy, cancer diagnosis and imaging, gene therapy and protein delivery, among others, have been thoroughly explored in recent years [4]. PLGA nanocomposites have also been explored for vaccination, brain targeting, cardiovascular and inflammatory diseases in addition to scaffolds for tissue engineering, as we will describe in this review. Nevertheless, research efforts are still required for the development and commercialization of PLGA nanocomposites for biomedical applications since they still present a number of limitations, such as poor drug loading, high burst release, high production costs and challenging scalability [4]. In this review we aim to explore recent advancements made in this area. It begins with a brief discussion on PLGA’s properties, approved PLGA-based medicines and an overview of the main production methods of PLGA MPs and NPs. The main focus of this review is then the combination of PLGA with INPs to form hybrid PLGA-based composites (nano-, micro- and macro-), and it is organized into three principal subjects: types of hybrid PLGA/inorganic-based materials, synthesis methods of these composites and their applications in the biomedical area, mainly in the form of carriers and scaffolds. This review aims to provide a backdrop for future research.

## 2. PLGA Physicochemical Properties and Synthesis Methods

### 2.1. PLGA Properties

As already mentioned, PLGA is produced by the catalyzed ring-opening copolymerization of the LA and GA units [7,8]. During polymerization the monomeric units are consecutively linked together through ester linkages, resulting in the formation of the PLGA copolymer [3,24]. PLGA is widely used in nanomedicine due to its biocompatibility and effective biodegradability, which occurs through the hydrolysis of the ester bonds of lactate and glycolate [2]. These monomers can then be metabolized via the Krebs cycle, yielding nontoxic byproducts (H_2_O and CO_2_) [7,25].

Different forms of PLGA can be obtained by varying the Poly(lactic acid):Poly(glycolic acid) (PLA:PGA) ratio during polymerization; for example, PLGA 50:50, which is frequently used in nanotechnology, has a composition of 50% lactate and 50% glycolate [26]. PLGA copolymers inherit the intrinsic properties of their constituent monomers, where the PLA:PGA ratio, along with the polymer molecular weight, influence their hydrophobicity, crystallinity, mechanical properties, size and biodegradation rate [24,27,28,29]. PGA is a crystalline hydrophilic polymer, while PLA is a stiff and more hydrophobic polymer; therefore, PLGA copolymers with a higher PLA content are less hydrophilic, tend to absorb less water and consequently present longer degradation times [30]. The degradation time can vary from several months to several years depending on the molecular weight (Mw) and copolymer ratio [4]. PLGA is soluble in a variety of solvents, including organic solvents such as chloroform, acetone, ethyl acetate and tetrahydrofuran [2,31]. All the above-mentioned features of PLGA have been proven to be useful for several applications; controlled drug release in particular.

PLGA can be block-polymerized with other copolymers, which can alter its behavior and physicochemical properties [32]. Diblock or triblock copolymers have been developed to meet the need for better carrier functionality, both in terms of the variety of drugs incorporated and administration methods [2]. Block copolymers of PEG (poly(ethylene glycol)) and PLGA are the most reported both in diblock (PLGA-PEG) [33] and triblock conformations (PLGA-PEG-PLGA or PEG-PLGA-PEG) [34,35]. The creation of PEG layers can reduce interactions with foreign molecules, increasing shelf stability in this way. However, it can also decrease drug encapsulation efficiencies. When compared with PLGA alone diblock copolymers have shown improved release kinetics [2]. The random combination of other polymers with PLGA can also be beneficial; for example, combining biodegradable photoluminescent polyester (BPLP) with PLGA will make the system suitable for photoluminescence imaging [36]. Thus, it is important to consider the final purpose of a system when choosing a polymer conformation [37].

Regarding the physical characteristics of nano-PLGA structures, they can be controlled by parameters specific to the production method employed. For example, the size of PLGA NPs can be determined to a certain extent by the concentration of polymer used for their synthesis [7]. Surface functionalization is another important aspect that allows a certain control over particles’ biocompatibility, biodegradation, blood half-life and, when applicable, targeting efficiency [7]. In fact, PEGylation has been shown to improve the pharmacokinetic properties of drugs encapsulated into PLGA composites [38]; coating PLGA NPs with biocompatible hydrophilic polymers (PEG or chitosan) can enhance stability and circulation time while diminishing toxicity [39,40].

In terms of biomedical applications, PLGA has been used in the clinic since 1989, being introduced mostly as microsphere formulations (Table 1); however, PLGA implants and nanocomposites are also a reality [41,42,43]. As mentioned above, PLGA is mostly utilized for drug delivery, having ~20 formulations that are FDA- and EMA-approved [43,44,45,46,47]. These are mainly administered by subcutaneous/intramuscular injections, yet they are versatile and present a plethora of applications. Table 1 presents a brief summary of these products as well as their respective production methods and applications [43,48,49,50].

Although many PLGA materials have already been commercialized, scientists are still trying to develop new methods of preparation that provide close control over their inherent characteristics. In the next section the most commonly used preparation methods for PLGA nanocomposites will be discussed.

### 2.2. PLGA MP/NP and Scaffold Preparation Methods

The physicochemical characteristics of PLGA particles can be controlled by manipulating specific parameters in their synthesis. It is of great importance to comprehend the different methods of PLGA particle preparation so that they can be used and manipulated to obtain optimized results. Although these are well-established methods each with different advantages, there are also limitations that should be considered in the preparation and development of PLGA materials. Common issues arise from mixing devices that usually do not allow for great batch-to-batch reproducibility, which leads to inconsistent particle behavior. The organic solvent used should be carefully chosen, since, similarly to the mixing device, it can create limitations in terms of emulsification, solvent extraction and particle homogeneity [32]. A wide range of techniques have been used for PLGA MP/NP synthesis, the most common one being the emulsification–solvent evaporation method, both single and double emulsion. Salting out, nanoprecipitation, emulsification–solvent diffusion and spray-drying are other common methods used in the synthesis of PLGA particles. For a better understanding these strategies will be discussed individually. Table 2 summarizes the main advantages and disadvantages of each method. A few techniques have also been reported for scaffold synthesis, with most of them resorting to molds to obtain a specific form. Electrospinning is one of the most used standard techniques for fibrous scaffold synthesis [51,52].

#### 2.2.1. Emulsification–Solvent Evaporation (ESE) Method

Single emulsion

Oil-in-water (O/W) emulsification is the most popular method for the preparation of PLGA particles, mainly when their intended loading is hydrophobic [53]. Following this methodology, appropriate amounts of PLGA polymer are first dissolved in an organic solvent (e.g., dichloromethane (DCM), chloroform or ethyl acetate). Subsequently, this organic solution is emulsified in an aqueous solution in the presence of a surfactant (e.g., polyvinyl alcohol, PVA) under continuous stirring. Afterwards, the organic solvent is allowed to evaporate, either by stirring at an adequate temperature or by applying a reduced pressure. The resultant particles are washed multiple times to remove polymer/surfactant residues, and they can then be freeze-dried [2,8] for long-term storage.

To load other components (drugs, imaging probes and NPs) into the PLGA matrix, they are usually co-dissolved in the organic solvent [2,54] before the emulsion is formed. Although O/W emulsions are the most popular emulsions created with this method oil-in-oil (O/O) emulsions are also used to encapsulate water-insoluble drugs. Here, the process is the same as in O/W emulsions, but the organic solvent phase is emulsified in a continuous oil phase (e.g., liquid paraffin or vegetable oil) [32].

Double emulsion

The double emulsion method is a more complex variation of the single emulsion method. The water-in-oil-in-water (W/O/W) emulsification method is the most used double emulsion method. In the single emulsion method the encapsulation efficiency (EE%) of hydrophilic compounds is very limited. In double emulsion methodologies the EE% and particle size are affected by the solvent used and the stirring rate [2]. For the W/O/W method an aqueous solution containing the materials to be encapsulated is emulsified in a PLGA-containing organic phase under vigorous stirring. Next, this water-in-oil (W/O) emulsion is added to a surfactant-containing aqueous solution under continuous stirring. Then, the organic solvent is allowed to evaporate via the same processes mentioned in the single emulsion method [24,32].

Even though W/O/W emulsions are by far the most used ones there are other double emulsion options available, such as water-in-oil-in-oil (W/O/O). Here, the second oil phase is made of an organic solvent that is miscible with the organic solvent of the first oil phase, but it is an antisolvent for PLGA [32].

#### 2.2.2. Salting-Out Method

In this method a solution of a water-miscible organic solvent containing PLGA and the compounds to be loaded is added to an aqueous phase containing a salting-out agent (e.g., calcium chloride) and a stabilizer (e.g., PVA) [24]. Under continuous stirring an O/W emulsion is first formed. Then, a large volume of water is added to the emulsion until the organic solvent diffuses into the aqueous phase, which leads to particle formation [2]. Finally, salting-out agents are removed by filtration and the particles are washed several times to remove the excess stabilizer. This method is ideal for high concentrations of polymer and for the encapsulation of heat-sensitive compounds [24,55].

#### 2.2.3. Emulsification–Solvent Diffusion (ESD) Method

This technique was first developed by Quintanar-Guerrero et al., and is a modification of the salting-out method [56,57]. Here, the organic solvent and water are mutually saturated at room temperature to attain a thermodynamic equilibrium. Afterwards, known amounts of polymer are dissolved into an organic solvent, and this solution is then emulsified in an aqueous solution containing a stabilizer (e.g., PVA) using a high-speed homogenizer [32,58,59]. Later, water is added to the O/W emulsion under regular stirring, which will allow the solvent to diffuse outwards from the internal phase. Then, the nanoprecipitation of the polymer occurs, leading to the formation of colloidal particles. Finally, the solvent can be removed either by evaporation or vacuum steam distillation [3,24,60]. This method presents high reproducibility and allows for a high EE% of hydrophobic drugs.

#### 2.2.4. Nanoprecipitation Method

The nanoprecipitation method, also known as the solvent displacement method, is a simple one-step process with high reproducibility that is mainly used to entrap hydrophobic drugs [7] in PLGA NPs. Here, the physicochemical characteristics of the resulting NPs depend on the PLGA monomers ratio and Mw, the solvents and the mixing rate [7,61]. In this method the polymer and cargo are dissolved in a water-miscible organic solvent (e.g., acetone, ethanol or acetonitrile). Then, this solution is added drop-by-drop into an aqueous solution containing a surfactant or emulsifier. The rapid organic solvent diffusion into water leads to the immediate formation of PLGA NPs. Finally, the organic solvent is removed under reduced pressures [7,24,32]. Modifications to this method have been made to adapt it to the encapsulation of hydrophilic drugs, for example, by replacing water with cottonseed oil and Tween-80 [7,62,63]. A two-step method has also been reported to encapsulate enzymes, where a first protein nanoprecipitation step is introduced, followed by a second nanoprecipitation of PLGA, resulting in protein-loaded PLGA NPs with a high EE% [64].

#### 2.2.5. Spray-Drying Method

The spray-drying technique is ideal for scaling-up the synthesis of PLGA particles. It is a rapid and convenient method with few processing parameters [2,8]. It sprays a W/O emulsion into a stream of hot air, which leads to the formation of particles. The solvent choice depends on the hydrophobicity of the cargo present in the W/O dispersion [8]. A major drawback for this method is the fact that the particles usually adhere to the inner walls of the spray-dryer [65].

**Table 2 ijms-23-02034-t002:** Advantages and disadvantages of PLGA particle synthesis methods.

Method	Advantages	Disadvantages
ESE [3,7,8,66]	-Encapsulates hydrophobic and hydrophilic agents. -Control of particle size. -Ease of scale-up.	-Time-consuming purification. -Needs heat or vacuum to remove solvent. -Biomacromolecule instability. -Instable W/O/W emulsions with poor drug EE%. -Batch-to-batch variability.
Salting out [7,24,29,32]	-Efficient encapsulation of heat-sensitive agents (proteins, DNA and RNA). -Low-energy mixing device.	-Time-consuming purification. -Not suitable for lipophilic drugs. -Use of large quantities of salting-out agents.
ESD [3,24,32,57]	-Simple and convenient. -Batch-to-batch reproducibility. -Ease of scale-up. -Monodisperse particle size. -High EE% (~70%).	-Leakage of water-soluble drugs into aqueous external phase, decreasing their EE%. -Large volumes of water to be removed.
Nanoprecipitation [7,8,24,32,67]	-Batch-to-batch reproducibility. -One-step process. -Smaller particle size. -Low-energy mixing device.	-Aggregation due to incomplete solvent removal. -Low EE% for hydrophilic drugs. -Negative effect of organic solvent on protein function.
Spray-drying [3,8,24,66,68,69]	-Encapsulates hydrophobic and hydrophilic agents. -Fast, convenient and few processing parameters. -Suitable for industrial scale-up.	-Adhesion of the particles to the spray-dryer wall. -Limited control of particle size.

#### 2.2.6. Electrospinning Method

There are several techniques used for scaffold synthesis, such as gas foaming [70], porogen leaching [71], phase separation [72] and twin-screw extrusion [73]. However, electrospinning [51,52] is by far the most popular one. The electrospinning technique has gathered more interest in recent years due to its versatility, simplicity and low cost in production. This technology has been highly useful in fabricating scaffolds for tissue engineering [51]. Briefly, it consists of a simple setup where a PLGA aqueous solution is kept in a syringe and injected through a needle by applying a high electrical voltage. Consequently, the needle becomes unstable and nanofiber jet spinning is achieved. The nanofiber is later collected on a conducting substrate [74]. The diameter of the fibers can be controlled by adjusting parameters, such as the voltage used, injection rate and collector type. This method can usually produce nanofiber sheets ranging from a few nanometers to several micrometers to be used as scaffolds [75].

## 3. Types and Synthesis of Hybrid PLGA Composite Materials

The development of hybrid PLGA materials represents a growing area in nanomedicine that has been employed as an efficient strategy to improve the structural and functional properties of PLGA. Although PLGA presents many advantages, its poor mechanical properties, the release of acidic byproducts, its hydrophobicity and suboptimal bioactivity are major bottlenecks for its applications [76,77]. To improve the properties of PLGA, its combination with various inorganic materials has been studied [16,51,78,79,80,81]. On top of this, inorganic nanomaterials bring into play interesting capabilities beyond those of PLGA, whether in the form of imaging/sensing properties or as fundamental physical properties. Furthermore, inorganic materials such as metallic and magnetic NPs usually present low stability and a tendency to agglomerate, which can be circumvented via their incorporation into PLGA matrices [16].

PLGA is often used for drug delivery purposes, where the use of both drugs and inorganic nanomaterials can potentiate the drug effect and equip the particles with new abilities (e.g., imaging). Gold-, silver-, iron-, manganese- and titanium-based nanomaterials, among others, are often utilized in combination with PLGA for this purpose. Additionally, PLGA is consistently used to fabricate scaffolds for tissue engineering applications since it offers control over their degradation as well as excellent properties, such as tensile strength and elastic modulus [82]. The fabrication of these scaffolds can be made directly from PLGA polymers or PLGA MPs. Nonetheless, the use of PLGA in tissue engineering is still limited due to its poor osteoconductivity, the release of acidic byproducts, the hydrophobicity of PLGA scaffolds, which alienates cell infiltration, and its suboptimal mechanical properties [76].

Several different combinations of PLGA copolymers with INPs have been explored and will be discussed in the following sections. A wide range of emulsion-based synthesis techniques are used to prepare these PLGA particle composites [8,83]. Each of these techniques have advantages and limitations (refer to Table 2), and they should be chosen according to the intrinsic features of the polymer, the physicochemical properties of the loaded drugs and INPs as well as the nature of the final application [84]. Most of the methods share common features of dispersing the PLGA polymer in an organic phase and mixing the solution with an antisolvent or aqueous phase. The solvent is usually removed by evaporation and/or extraction [8,32,85]. Depending on the chemical nature of the materials to be encapsulated, they are usually added either to the organic or to the aqueous phase.

### 3.1. PLGA/Plasmonic Nanocomposites

Plasmonic materials have gathered great research attention in recent years due to their tunable optical properties, which are achieved through the exploitation of their surface plasmon resonance (SPR) effects [86]. Gold and silver are the most commonly used plasmonic materials. The wavelengths at which they interact with light can be tuned by changing their size and shape, and in this way bespoke materials can be prepared to match their intended application [87]. Figure 1a presents two different PLGA/plasmonic nanocomposites.

Gold NPs (AuNPs) in the form of spherical nanoparticles, nanorods or nanocapsules are the most widely used plasmonic NPs in biomedical applications [88]. They present a localized SPR property, which allows the NPs to absorb light (in the visible to near infra-red range) and transform it into heat, ideal for thermal therapies [89]. AuNPs are frequently used due to their high chemical stability, easy functionalization, low toxicity and high optical absorption as well as photoacoustic (PA) signal [90,91]. Additionally, they have been proposed as a bone morphogenic substance, presenting not only inhibition properties to the formation of osteoclasts but also supporting osteoblast differentiation [92,93]. AuNPs have been widely studied in combination with PLGA, mainly for theranostic purposes, combining photothermal therapy with X-ray or PA imaging [16,78,91]. Several strategies to incorporate AuNPs into PLGA composites have been described, from encapsulation into the polymer core [91,94,95] to the decoration of and/or growth on their polymeric surface [78,96,97] to the coating of scaffolds [52].

Silver NPs (Ag NPs) are also known for their localized SPR, which allows control over their optical absorption and temperature profile [98]. However, in nanomedicine Ag NPs are mostly used due to their nontoxicity, biocompatibility, antimicrobial as well as antioxidant activity and ability to enhance the antibacterial activity of different drugs [99,100]. Several strategies have been presented to potentiate the antibacterial efficacy of the organic–silver NP combination, both as PLGA particles [19] and as functionalized scaffolds [51].

#### 3.1.1. PLGA Materials with Plasmonic NPs in the Core

Fazio et al. [95] synthesized a PEG-PLGA copolymer nanocomposite loaded with AuNPs and silibinin (SLB), where the AuNPs were produced by laser ablation and immediately embedded into a previously prepared PEG-PLGA copolymer. After the bespoke preparation of the PEG-PLGA copolymer, the ESD method was followed for the preparation of the final nanocomposite: the PEG-PLGA copolymer and SLB were dissolved together in DCM, and this solution was immediately added into a AuNP solution and ultrasonicated, forming a W/O emulsion. Next, an aqueous solution was added to the W/O emulsion, which induced organic solvent diffusion outwards from the internal phase. The final emulsion was centrifuged, eliminating the low-molecular-weight polymer. The resulting SLB-loaded PEG-PLGA_Au nanocomposite presented a porous spherical form that was 200 nm in size. The encapsulated AuNPs ranged from 5–50 nm in size and enabled the use of this nanocomposite for photothermal therapy (PTT) and light-controlled drug release [95].

In another study by Deng et al. [94] small AuNPs (3–5 nm) were encapsulated together with the photosensitizer verteporfin (VP) into a PLGA matrix, at varying gold:VP ratios. These PLGA-AuNPs were synthesized by a single ESE method, obtaining spherical NPs approximately of 100 nm diameter. These NPs were suitable for cancer therapy through photodynamic therapy (PDT). Wang et al. [91] synthesized NPs containing silica-coated AuNPs and perfluorohexane (PFH) liquid in the core via the double emulsion method. The NPs were stabilized by a PLGA shell where the hydrophobic drug paclitaxel (PTX) and a fluorescent dye were incorporated (Figure 1a right). The AuNPs were produced via a single-phase aqueous reduction of tetrachloraunic acid by sodium citrate, further coated with silica and fluorinated [101,102,103]. PLGA particles were then prepared using a W/O/O double emulsion (Figure 2a) [91]. The resulting nanocomposites were spherical and had an approximate size of 550 nm. Due to their components, the particles are able to work as PA and fluorescent imaging agents as well as antitumoral agents.

#### 3.1.2. PLGA Materials with Plasmonic NPs at the Surface

Hao et al. used a modified emulsification solvent evaporation method [104] to synthesize PLGA-based NPs loaded with docetaxel (DTX), that were modified with polyethyleneimine (PEI) and a shell of AuNPs with the presence of the glioma-targeting peptide angiopep-2 (ANG/GS/PLGA/DTX NPs). First, PEI-modified PLGA/DTX NPs were prepared. Then, the incorporation of gold onto the surface of the NPs was performed. Negatively charged AuNPs were easily attracted to the PEI-PLGA NPs for subsequent growth of the gold overlays. Gold nanoseeds were formed on the surface of PLGA/DTX NPs in two steps: (i) the reduction of hydrogen tetrachloroaurate hydrate with sodium borohydride and then (ii) the reduction of hydrogen tetrachloroaurate hydrate with ascorbic acid, which allowed for a slow growth of the Au nanoseeds and the consequent formation of gold nanoshells. The final targeted particles were obtained via the incubation of the nanocomposites with angiopep-2 and HS-PEG2000 [78]. The obtained particles were spherical and had a size of approximately 200 nm. The nanocomposites were intended to be used as chemophotothermal agents (DTX plus PTT) and X-ray imaging agents in cancer therapy [78].

Likewise, Song et al. developed plasmonic composite PLGA vesicles [96] approximately 60 nm in size, where gold nanorods (AuNRs) were embedded in the vesicle shell formed by PLGA, with PEG molecules extending to the interior and exterior of the vesicle, providing stability. Inorganic cetyltrimethylammonium bromide (CTAB)-modified AuNRs were synthesized via a one-spot seedless method [105]. Then, for the synthesis of amphiphilic AuNRs attached with PEG and PLGA a “grafting to” reaction was performed [106,107,108]. This reaction happens by mixing thiolated PEG and PLGA (PEG-SH and PLGA-SH) with AuNR, where the polymers are “grafted” to the AuNRs’ surface by forming Au-S covalent bonds [96]. Subsequently, the AuNR@PEG/PLGA vesicles were prepared using the emulsification solvent evaporation method. PLGA forms a vesicular shell embedded with AuNRs and PEG extended inside and outside the PLGA vesicle, stabilizing it in aqueous solution and preventing aggregation. The vesicle size can be controlled to some extend by controlling the concentration of AuNRs@PEG/PLGA in the organic phase. These AuNRs@PEG/PLGA nanocomposites were designed as theranostic agents using PA image-guided PTT [96].

In another study, Topete et al. [97] developed a multifunctional PLGA-based nanocomposite loaded with the anticancer drug doxorubicin (DOX), covered with a porous Au-branched shell (BGNSH) and functionalized with a human serum albumin/indocyanine green/folic acid complex (HSA-ISG-FA). The authors utilized the nanoprecipitation method to form these nanostructures. To form the Au shell around the PLGA core PLGA NPs were incubated with chitosan in order to invert the NPs’ surface charge from negative to positive [109]. Next, the NPs were incubated with a Au seed solution followed by a Au growth solution to form the shell. Finally, this solution was reduced with ascorbic acid [97]. The resultant NPs were spherical and had an approximate size of 125 nm (Figure 2b). All the elements of this nanocomposite synergistically combined to obtain extended therapeutic effects and also serve as contrast agents for optical imaging due to the NPs’ fluorescence [97].

Takahashi et al. prepared silver-decorated PLGA NPs (Ag PLGA) with an approximate size of 300 nm [19] (Figure 1a left) in a different manner: first, PLGA NPs were obtained by the ESD method, similarly to the above-mentioned studies; afterwards, a silver modification was performed. PLGA NPs were dispersed in water and added to an aqueous AgNO_3_ solution. This solution was then mixed with NaBH_4_. The resulting solution was freeze-dried and stored for further use [19]. Given the antimicrobial activity of silver, these particles were designed to act as a DDS against biofilm infections.

In a different approach Lee et al. developed PLGA nanofibrous scaffolds that were chemically coated with AuNPs by using electrospinning [52]. The PLGA was directly synthesized and functionalized with thiol ends to efficiently bind AuNPs (30 nm size). For the binding with AuNPs, the thiolated PLGA sheets prepared by electrospinning were immersed in AuNP solutions at different concentrations. AuNPs promote osteoblast differentiation and allow for the attachment of drugs to the system. Therefore, the synthesized composite scaffolds are promising tools as controlled release scaffolds for bone tissue regeneration.

In a different study by Almajhdi et al. PLGA nanofiber sheets were functionalized with Ag NPs at different percentages to evaluate their influence in the fibers’ performance [51]. To form the Ag NPs AgNO_3_ at different ratios was dissolved in tetrahydrofuran or dimethylformamide and reduced with PEG, which also served as a stabilizer. This solution was stirred for 30 min before PLGA was added to the solution. This mixture would later be electrospun to form PLGA@Ag NP nanofibers. The scaffolds synthesized by electrospinning were highly porous materials and had nanofiber diameters of around 100–200 nm, while the size of the Ag NPs went from 5–10 nm. Owing to the antibacterial and ROS (reactive oxygen species)-generating activity of the Ag NPs the scaffolds are ideal candidates for tissue engineering and suitable for anticancer as well as antibiotic drug delivery.

### 3.2. PLGA/Magnetic Nanocomposites

During recent decades nanotechnology has enabled the production of bespoke magnetic NPs (MNPs), characterized by a tailored response to an applied magnetic field. MNPs have been used in the clinic since the late 1990s as contrast agents (CAs) in magnetic resonance imaging (MRI) [110]. MNPs can also be used to generate heat under alternating magnetic fields (AMFs), useful for the direct ablation of tumors through magnetic hyperthermia and/or for the controlled delivery of drugs [111,112,113,114,115,116]. Additionally, some MNPs showcase a responsive behavior to changes in pH and redox states [117,118]. Figure 1b presents two different MNPs combined with PLGA to form functional magnetic composites.

Iron-oxide-based NPs (Fe_x_O_y_ NPs) are one of the most researched structures in nanomedicine due to their relevant physicochemical properties and magnetic properties, facile synthesis and versatility of applications [119]. For most of the proposed applications in biomedicine the particles perform best below a critical diameter (<20 nm), being called superparamagnetic iron oxide nanoparticles (SPIONs) [119,120], where their magnetic properties are within the superparamagnetic regime.

Several strategies have been developed to encapsulate magnetic NPs onto PLGA MPs and NPs with the goal of obtaining multifunctional composites with diagnostic (mostly via PA or MRI [121,122,123]) and/or therapeutic purposes [36,121,122,123,124,125,126,127,128,129].

Sun et al. combined SPIONs with PLGA in such a way that the magnetic NPs were embedded in a polymer shell while the core was filled with liquid [124]. Here, a double ESE method (W/O/W) was used to synthesize microcapsules, which presented a nonuniform size of approximately 900 nm. The particles were designed as theranostic agents through the use of ultrasound (US) and MR imaging as well as to enhance the therapeutic efficiency of high-intensity focused ultrasound (HIFU) [124]. Based on a previous study [130], Schleich et al. [126] synthesized PEGylated PLGA-based NPs loaded with SPIONs and PTX or DOX (Figure 1b right) via the ESD method (Figure 2c). The SPIONs were first synthesized by a classical coprecipitation method of iron salts in an alkaline medium [131]. To fabricate the PLGA NPs, PLGA, PLGA-PEG, poly(Ɛ-caprolactone-b-ethylene glycol) (PCL-PEG) and the drugs were dissolved in a SPION DCM solution. This organic phase was emulsified with a PVA aqueous solution. Then, this O/W emulsion was added dropwise to an aqueous solution containing PVA; DCM was subsequently evaporated [126]. The obtained PLGA NPs were spherical, and when containing PTX and DOX presented an approximate size of 250 nm and 290 nm, respectively. The repulsive forces between PEG molecules provided higher stability to the NPs in serum fluids, making them ideal for drug delivery applications [132,133]. The particles were also capable of producing *T*_2_ contrast in MRI. The contrast observed in MR images is the result of local variations in longitudinal (*T*_1_) and transverse (*T*_2_) relaxation times of water molecules in regions adjacent to the injected particles. Likewise, Ruggiero et al. also synthesized PLGA NPs containing Fe_3_O_4_ NPs and PTX using the method described below [125]. A classical approach was used to fabricate Fe_3_O_4_ NPs via coprecipitation from a solution of iron(II) and iron (III) chloride in water using ammonium hydroxide as a base and later adding oleic acid [125,134,135]. In the reviewed studies the INPs and, when in cases where applicable, the drug were added into the organic phase of the single emulsion method [79,122,123,125]. For example, in a study by Ruggiero et al. a mixture of the produced Fe_3_O_4_ NPs, PLGA and PTX was constructed in chloroform [125]. This mixture was later added in a dropwise manner to an aqueous solution of PVA and sonicated. A rotatory evaporator was then used to evaporate the solvent from the obtained emulsion [125]. In this way PTX-loaded PLGA-Fe-NPs were obtained. The particle size was between 120–160 nm and their magnetic core size was around 15–20 nm. The PLGA-Fe nanocomposites were ideal for magnetic-hyperthermia-triggered drug release and could also be used as MRI contrast agents.

All reported studies where Fe_3_O_4_ NPs were encapsulated into PLGA propose very similar synthetic protocols both for the INPs and for the PLGA particles. Several studies used a thermal decomposition method [136,137] to synthesize iron NPs by heating up a mixture of an iron–oleate complex in the presence of oleic acid, with a high boiling point ether serving as a solvent [79,122,123]. This method usually produces high-quality and very monodisperse nanocrystals but is more cumbersome than, for example, co-precipitation. The above-mentioned studies used the single emulsion technique in a very similar way to Ruggiero et al., with relevant differences being the cargo, solvents used (methylene chloride [122,123] or DCM [79]) and evaporation method (stirring for 3 h [122,123] or agitating overnight [79]). Moreover, Nkansah et al. [123] used this same method to synthesize both PLGA MPs and NPs only by varying the %(*w*/*v*) of PVA in aqueous solutions and the homogenization method used.

ESE W/O/W emulsions are also very frequently used to prepare magnetic-PLGA composites and have different variations. The MNPs can be included in the first aqueous phase and emulsified with a PLGA-containing organic phase. Then, a PVA-containing second aqueous phase is added to the emulsion and emulsified, forming a W/O/W emulsion. Lastly, the emulsion is left stirring for several hours to evaporate the organic solvent [80,121,127]. Lee et al. [127] developed a more complex version of a multifunctional PLGA nanocomposite with a magnetite core using this approach. In this study, PLGA was linked with methoxy poly (ethylene glycol) (mPEG) and/or chlorin e6 (Ce6) through the Steglich esterification method [138]. Magnetite NPs were dispersed in water, making the first aqueous phase. PLGA-mPEG and/or PLGA-Ce6 were added to DCM, making the organic phase. The two phases were mixed together with vigorous vortexing, producing an emulsion. This emulsion was added to a second aqueous phase containing PVA and NaCl and emulsified. Finally, the solution was stirred to evaporate the organic solvent and collected by centrifugation [127]. The conjugation with mPEG was used to provide higher stability to the nanocomposites in biological fluids; with Ce6 the goal was to obtain fluorescent in vivo images and perform PDT tumor therapy, damaging tumor cells through ROS generation [139,140]. Due to the magnetic core it was also possible to perform MRI diagnoses.

Some researchers also proposed to include the iron-based NPs on the surface of PLGA NPs, or embedded into a polymer shell. Fang et al. [129] produced magnetic PLGA microspheres loaded with DOX (DOX-MMS), with an approximate size of 2.4 μm. In this example the drug was encapsulated in the core, and the iron NPs, in this case γ-Fe_2_O_3_, were electrostatically assembled on the microsphere’s surface, which was precoated with PEI. The magnetic NPs were synthesized by coprecipitation and coated dimercaptosuccinic acid [141], while the polymeric particles were prepared via a modified W/O/W emulsion. The main significant difference between this and the above-mentioned double emulsion is that the inorganic material is not included when forming the W/O/W emulsion. Instead, the incorporation of the magnetic NPs to the PLGA microspheres happens as a postsynthetic modification through electrostatic deposition with the aid of a PEI coating [129]. The authors propose that in this construction the resulting microcomposites are more responsive and that under an external magnetic field the drug release will be enhanced, showing that the particles are ideal for magnetic-hyperthermia-induced drug release and chemothermal therapy.

In other studies where the double emulsion method was also used the MNPs were included together with PLGA in the organic phase of the W/O/W emulsions. Zhang et al. [36], for example, incorporated SPIONs into a polymer shell made of two different polymers: PLGA and BPLP. The photoluminescent (from the BPLP) nanocapsules where coated with PEG to give them an enhanced stealth effect, and BSA protein was encapsulated into their core. SPIONs coated with oleic acid were first synthesized by a microwave-assisted thermal decomposition method and then later incorporated into the polymer NPs’ shell [142]. The PLGA-BPLP copolymer was synthesized by using BPLP as a macroinitiator that reacts with lactate and glycolate via ring-opening polymerization. PLGA-PEG was also used to obtain the PEGylated nanocapsules. For the synthesis of these nanocapsules an aqueous phase containing the encapsulant (bovine serum albumin, BSA) was emulsified in an organic phase containing SPIONs and different proportions of PLGA. Next, this W/O emulsion was emulsified with a second water phase containing PVA, and the second W/O/W was formed by sonication [36]. This DDS was proposed as a theranostic platform for MRI/photoluminescence dual-mode imaging and drug delivery. In a different example, Saengruengrit et al. [128] also developed core–shell magnetic PLGA nanocomposites, in which SPIONs were embedded in the polymer shell and the core was composed of an aqueous phase containing BSA (as a protein antigen model). Oleic-acid-coated SPIONs were synthesized by a classic thermal decomposition method [136]. SPIONs were also included in the PLGA NP shell and BSA was encapsulated in the particles’ core, it being the case that the synthesis by double emulsion was performed in the same manner [128]. To encapsulate BSA the particles were separated into two size groups, one with 500 nm and another with 300 nm, to study the effect of NP size on immune modulation. The composites were envisioned as vehicles for antigen delivery to stimulate an adaptive immune response when combined with an alternating magnetic field.

Another widely studied type of MNPs is manganese-oxide-based NPs. These present a strong paramagnetic character, and therefore these particles do not retain any magnetization in the absence of an external magnetic field [118]. Mn NPs are mainly used as *T_1_*-MRI contrast enhancers [143,144,145].

Bennewitz et al. [146] developed pH-sensitive PLGA NPs and MPs encapsulating MnO nanocrystals of 15–20 nm size (Figure 1b left). A single ESE with minor modifications was used to synthesize these PLGA NPs. The nanocrystals were synthesized via the controlled thermal decomposition of manganese(II) acetylacetonate in benzyl ether and oleic acid. Afterwards, the PLGA NPs were fabricated as follows: for the organic phase PLGA was dissolved in methylene chloride, and dried MnO NPs were added to this solution. The organic phase was added dropwise to an aqueous phase containing PVA as a stabilizer. This mixture was vortexed and sonicated to obtain a O/W emulsion. This emulsion was added to another aqueous PVA solution under rapid stirring. The resulting nanocomposites were left to stir to evaporate the residual organic solvent. The obtained NP and MP composites presented average diameters of 140 nm and 1.7 µm, respectively. This system is intended to be used for molecular and cellular MRI imaging [146].

### 3.3. Other PLGA/Inorganic-NP-Based Composites

Many other inorganic NPs have also been explored in combination with PLGA polymeric materials. Zinc-, cerium-, carbon-, copper-, hydroxyapatite (HAp)- and titanium-based nanoparticles are examples of different inorganic materials that have been investigated in combination with PLGA and will be addressed in this section. Zinc oxide (ZnO) presents great optical, catalytic and semiconducting properties; additionally, it can be used for a plethora of biomedical applications due to its antimicrobial activity, biocompatibility, chemical stability and nontoxicity [99,100,147]. ZnO NPs are also able to generate ROS, which help in their anticancer activity in addition to their antibacterial activity against both Gram-positive and Gram-negative bacteria [148,149]. Other interesting INPs are cerium oxide NPs (nanoceria, CeO_2_), which display a recyclable antioxidative activity, being able to act as ROS-scavenging NPs due to their ability to regenerate their oxidation state under various environmental conditions [150]. Their antioxidative activity mimics the activity of superoxide dismutase (SOD) and catalase through the conversion of superoxide radicals and hydrogen peroxide into oxygen and water [151]. Carbon nanotubes (CNTs) are another example of a nanomaterial widely used in a variety of applications. They have high surface area, low density, high mechanical integrity and exhibit a high mechanical modulus (0.2–1 TPa) [152]. These properties make CNTs ideal nanomaterials to be incorporated into scaffolds, for example. Copper oxide (CuO) NPs have gathered great research attention since copper is one of the most used metals and presents useful optical, electrical and medicinal properties [153]. It is predominantly used in nanomedicine due to its antimicrobial activity against a range of bacterial strains, which mainly occurs through the release of Cu^2+^ ions [154,155]. Titanium dioxide (TiO_2_) is a very important material used in the fabrication of bone implants and has received great attention due to its biocompatibility and appealing mechanical properties [156]. Scaffolds incorporating TiO_2_ NPs showcase great physical properties, such as corrosion resistance, low weight and low toxicity, but also great bioactive properties, such as enhanced cell proliferation and cell adhesion [157,158,159,160,161]. Inorganic HAp NPs are another important class of NPs that are widely used in combination with PLGA [162,163]. HAp is a strong inorganic material that is naturally present in the bone, considered to be a vital component for scaffolds due to its excellent biocompatibility, bioactivity, lack of immunogenicity and osteoconductivity [77,164].

Stankovic et al. [165] prepared PLGA NPs loaded with inorganic ZnO NPs (Figure 1d) via a solvent/nonsolvent method. ZnO NPs were prepared via a modified microwave-assisted synthesis [166], while PLGA NPs were prepared by the aforementioned solvent/nonsolvent method. For the preparation of the final composite materials PLGA NPs were dissolved in acetone (or ethyl acetate), and subsequently a solution of ZnO in acetone was added dropwise under constant homogenization. Ethanol was then added to the mixture to form a precipitate, and this suspension was slowly poured into an aqueous solution containing polyvinylpyrrolidone (PVP) as a stabilizer. The resulting particles were spherical and uniform, presenting a size of approximately 200 nm. This system can be used for different biomedical applications, such as drug delivery and treatment of bacterial infections [165].

In a different study, Mehta et al. [167] fabricated PLGA MPs containing nanoceria and ROS-scavenging enzymes, either catalase or SOD, using a previously reported one-stage method [168]. These PLGA MPs are expected to showcase a synergistic effect with the enzymes and were synthesized via a standard double emulsion method [80], where the encapsulants (nanoceria and SOD) were dispersed together with PLGA in the organic phase [167]. The particles were uniform and presented a mean diameter of 800 nm. The synthesized PLGA microcomposites can be used to protect cells from oxidative stress. Singh et al. [80] also synthesized PLGA MPs encapsulating nanoceria (CNP-PLGA), with an average size of 60 µm and a porous surface, using the same standard double emulsion method (Figure 1d). They also incorporated nanoceria into PLGA scaffolds to be used in tissue engineering. The fabrication of PLGA scaffolds can be performed with a wide variety of techniques and depends on the desired scaffold structure and application. In this study the scaffolds were constructed by solution casting and rapid evaporation under reduced pressure. Since the nanoceria particles mimic the activity of SOD, their controlled and sustained release from PLGA MPs can make them useful for a variety of biomedical applications, such as mitigating damage from radiation and bacterial infection [80].

Other PLGA scaffolds combined with different INPs have been widely studied. Mikael et al. [169] combined multiwalled carbon nanotubes (MWCNTs) and PLGA to fabricate a mechanically enhanced and biodegradable composite scaffold from PLGA-MWCNT microspheres. PLGA-MWCNT microspheres were first synthesized using the single emulsion method (O/W). Afterwards, these MPs, in a size range of 425–600 µm, were used to fabricate circular scaffolds via thermal sintering. The particles were packed into a steel mold that was heated at 95 °C for 1 h, and then were cooled down and stored in a desiccator [169]. These scaffolds showed enhanced mechanical properties, indicated to be used in bone tissue engineering.

In another study Haider et al. [155] synthesized hybrid nanofiber PLGA scaffolds compositing CuO NPs by electrospinning (Figure 3a). For the synthesis of CuO NPs Cu powder was added to distilled water and sonicated. The reaction mixture was transferred into a glass bottle, sealed and autoclaved. CuO NPs were retrieved by centrifugation. To synthesize the scaffolds a PLGA/CuO solution was prepared and put into a syringe to be electrospun. The nanofibers were smooth and uniform, and their average diameter was around 550 nm. These scaffolds were proposed for antibacterial purposes due to the antimicrobial activity of CuO NPs.

In a more recent study Pelaseyed et al. [170] fabricated PLGA nanocomposite scaffolds functionalized with commercially available titanium oxide (TiO_2_) NPs at different ratios. These scaffolds were fabricated by an air–liquid foaming process that uses nucleation to create gas bubbles, which results in a porous microstructure. The functionalization with TiO_2_ offers to improve the mechanical and bioactive properties of PLGA [157]. The scaffolds were highly porous and their pore size was reduced with higher contents of TiO_2_, resulting in an improvement in the mechanical properties of the scaffolds, ideal for bone regenerative engineering.

Sheikh et al. [77] fabricated hybrid PLGA–silk fibroin composite scaffolds combined with HAp NPs (PLGA–silk–HAp scaffolds). According to the authors, both silk fibroin and HAp NPs cannot be used alone: silk fibroin has a slow degradation rate, making it difficult for the scaffold to be replaced by tissue; HAp NPs present a free needle-like particulate nature, which results in frail films that are inadequate to be used as bone grafts [77,171]. Hence, their combination with PLGA can improve the mechanical properties, biodegradability, hydrophilicity and osteoconductivity of the scaffolds, making them suitable for the intended application in bone tissue regeneration. Porous PLGA–silk–HAp scaffolds were fabricated by salt leaching and vacuum mixing. First, pristine PLGA scaffolds were prepared as follows: PLGA pellets were dissolved in methyl chloride, mixed with NaCl particles (180–250 mm) in a ratio of 1:9 and poured into silicon molds. The molds were pressed for 24 h at room temperature. Afterwards, the samples were kept in a vacuum oven at 25 °C for 1 week to remove residual solvents. A salt-leaching technique was carried out to porosify the PLGA by using distilled water. Finally, the samples were dried at room temperature to form porous PLGA scaffolds. Afterwards, to obtain the final PLGA–silk–HAp scaffolds, an aqueous solution of HAp NPs was sonicated and added to a 4% silk solution, blended and poured onto the pristine PLGA scaffolds. Vacuum mixing was performed by turning the vacuum pump ON and OFF; in this way the scaffold pores were filled with the solution. In a related study, Selvaraju et al. [172] synthesized collagen/PLGA-based composite scaffolds by also incorporating inorganic HAp NPs, using a different approach. The HAp nanopowder was synthesized through a modified chemical precipitation method [173]. Initially, an aqueous solution of H_3_PO_4_ was added dropwise to a Ca(OH)_2_ solution. The pH of the final solution was adjusted via the addition of NH_4_OH. Finally, the filtered cakes were oven-dried to remove free water. To fabricate the scaffolds a PLGA/HAp composite was first synthesized, where an in situ preparation of NPs in a polymeric matrix was performed. For this synthesis the PLGA copolymer was prepared from scratch using the individual monomers in its constitution. While this method is not a common technique for the synthesis of PLGA composites, the HAp NPs can grow inside the polymer matrix, reducing aggregation and maintaining a good spatial distribution. The in situ polymerization was conducted by mixing HAp nanopowder with monomers, d,l lactide and glycolide under a N_2_ atmosphere. An initiator of the polymerization, stannous octoate, was added, and the reaction was performed for 12 h. Collagen was then dissolved in an acetate buffer and mixed with the PLGA/HAp composite. For the scaffold fabrication a vacuum drying technique was used [172]. The incorporation of HAp allows for an improvement in the composite physicochemical properties and stability, making it possible to form a scaffold of a defined pore structure. In this way it is possible to use this composite as a scaffold for regenerative engineering.

### 3.4. PLGA Composites Incorporating More Than One Inorganic Nanomaterial

Composite PLGA-based materials combining more than one inorganic nanomaterial have also been investigated [79,174,175]. Numerous types of combinations and designs can be made, e.g., including both INPs in the PLGA shell, only in the core or both in the core and shell. When combining together different types of INPs it is important to take into account their physicochemical properties to obtain a viable particle design. The concentrations of each material need to be thoughtfully optimized in a way that their abilities are not compromised and can perform synergistically [73,176]. With the use of these combined inorganic materials, it is possible to incorporate additional advantages into the system that will be able to perform multiple functions.

Ye et al. [79] combined inorganic imaging agents of manganese-doped zinc sulfide (Mn:ZnS) quantum dots (QDs, optical imaging) and SPIONs (MRI) by encapsulating them into PLGA vesicles together with the anticancer drug busulfan (PLGA-SPION-Mn:ZnS). The SPIONs were synthesized via thermal decomposition [136] and the QDs were synthesized via a nucleation-doping strategy [177]. These composites were fabricated via the single ESE method, where the INPs were included in the organic phase and worked as DDS theranostic agents.

In another study by Wang et al. [175] PLGA NPs were loaded with AuNRs coated with CTAB and the anticancer drug DTX, and then further coated with manganese dioxide (MnO_2_) ultrathin nanofilms (PLGA/AuNR/DTX@MnO_2_ NPs). These particles are illustrated in Figure 1c. For their synthesis AuNRs were synthetized through the CTAB-induced seed-mediated growth method [178,179]. AuNRs and DTX-loaded PLGA NPs were prepared using the simple ESE method. Lastly, to obtain the final MnO_2_-nanofilm-coated PLGA nanocomposites a 2-(*N*-morpholino) ethanesulfonic acid (MES) oxidation–reduction method was employed [180]. The final nanocomposites presented a spherical form with a rough surface due to the MnO_2_ shell, and their average size was around 280 nm (Figure 3b). These PLGA/AuNR/DTX@MnO_2_ structures were promising nanocomposites for theranostic applications, being able to perform dual-mode diagnosis (MRI from the Mn and X-ray CT from the Au) and radiofrequency-induced hyperthermia combined with chemotherapy [175].

Li et al. [174] developed core–shell nanocomposites based on PLGA copolymers using a stabilizer-free method. Here, the PLGA was loaded with PTX, resulting in spherical, ~200 nm size NPs that were further modified with the surface growth of a silver/gold nanoshell around the PLGA core (PLGA@Ag-AuNPs). Although two different materials are used this case differs from the other examples since both the materials have the same function. The particles’ nanoshell allows for a greatly improved surface-enhanced Raman spectroscopy (SERS) signal, making this platform very attractive for biodetection and controlled drug release, serving as a promising theranostic tool.

## 4. Applications

PLGA copolymers tailored in combination with INPs help to broaden the range of applications in biomedicine for which PLGA can be used. Some of the most explored uses for these hybrid polymer composites are therapy [97,129,181,182] (e.g., PTT, chemotherapy), diagnostics [121,122,123,146] (e.g., MRI, X-ray and PA), theranostics [78,79,96,175] and tissue engineering [77,155,169,172]. In this section we will briefly discuss several representative studies of each of these biomedical applications, each presenting different combinations of PLGA with INPs, in the form of MPs, NPs and macroscopic structures. Different targeting ligands and drugs are also considered.

### 4.1. Therapy

Several PLGA-only nanoparticles are already used in the clinic for the treatment of a variety of conditions (see Table 1). The combination of biocompatible, nontoxic and biodegradable PLGA nanoparticles with INPs can improve and expand the functionalities of each individual nanoparticulated counterpart in this area.

A good example is the combination of PLGA nanoparticles with gold nanoparticles for cancer treatment. In the work by Fazio et al. [95], the authors used a PEG-PLGA copolymer nanoplatform incorporating AuNPs and SLB for cancer therapy (Figure 4a). In this example, AuNPs absorb energy and transform it into heat, resulting in hyperthermal cancer therapy and photothermal drug delivery. Laser-irradiated NPs showed higher drug release when compared to nonirradiated samples, with a 75% relative release increment in the first 5 h of drug release (Figure 4b), proving that these NPs have great potential for biomedical applications. Nevertheless, in vitro and in vivo studies are still required to assess their biocompatibility and real anticancer potential. In a similar example, Deng et al. [94] incorporated AuNPs and the photosensitizer verteprofin (VP) in PLGA nanocomposites (PLGA@AuNPs-VP). In this case, AuNPs in combination with a photosensitizer can be used for PDT. In fact, the nanocomposite decreased the viability of pancreatic tumor cells after laser exposure (31% of live cells treated with PLGA-VP-AuNPs, compared to 44% of live cells treated only with PLGA-VP nanoparticles, Figure 4c), with an increase in the production of singlet oxygen, helping in the killing of tumor cells.

Besides the use of AuNPs, iron oxide NPs have become a suitable partner for PLGA-based particles. Due to their superparamagnetic properties iron oxide nanoparticles can respond to an AMF, resulting in the production of heat in a process called magnetic hyperthermia. Fang et al. [129] synthesized DOX-containing PLGA microspheres coated with Fe_3_O_4_ NPs (DOX-MMS). In vitro and in vivo studies demonstrated that DOX release from the microsphere increased after exposure to an AMF, resulting in the killing of tumor cells (81.6% of viable 4T1 cells when treated with DOX-MMS versus 10% of viable cells when treated with DOX-MMS and exposed to an AMF). In vivo studies using 4T1 tumor models have shown that mice treated with DOX-MMS and exposed to an AMF had reduced tumor growth (3.4 ± 0.6-fold, compared to 6.2 ± 0.8-fold in the control group and 4.0 ± 0.8-fold in the group treated with DOX-MMS without an AMF). Interestingly, PLGA coated with Fe_3_O_4_ NPs without a drug (MMS) also resulted in impaired tumor growth after exposure to an AMF, with tumors being smaller than the DOX-MMS without an AMF group, showing that the effect of the final formulation is due to a combination of magnetic hyperthermia (thermal therapy) and enhanced drug release. Fe_3_O_4_/PLGA nanoparticles can also be helpful for immunotherapy purposes, as shown by Saengruengrit et al. [128]. Under an AMF, Fe_3_O_4_/PLGA nanoparticles encapsulating BSA were internalized by macrophages and bone-marrow-derived dendritic cells. Furthermore, the polymeric particles stimulated the differentiation of immature dendritic cells in vitro, promoting the expression of proinflammatory surface markers and cytokines. In principle, these hybrid PLGA nanocomposites can be used for vaccination purposes since BSA could be replaced by any real antigen of interest.

Besides applications in the oncology arena, PLGA composites were also investigated for the treatment of infections. Stankovic et al. [165] and others [19,183] have shown that PLGA NPs with nano-ZnO and organic-Ag hybrid materials display a bacteriostatic effect against a wide range of Gram-positive and Gram-negative bacteria, as well as against the yeast Candida albicans. Therefore, the combination of Ag NPs with PLGA polymers can be very beneficial in this arena. Takahashi et al. [19] tested the efficacy of PLGA-Ag NPs against Staphylococcus epidermidis biofilms covered with a thick film of extracellular polymeric substances (EPS). The antibacterial activity of Ag salts in general and Ag NPs in particular has been well-characterized [155,184]. The NPs were able to adhere to the biofilm, and as PLGA degrades the Ag NPs promote the dissolution of the EPS film. The bacteria were damaged by the interaction with Ag NPs and voids were induced in the biofilms. Additionally, Ag NPs can cause the separation of the bacterial cell wall, leading to their death. Furthermore, the interaction of silver ions from the degradation of the Ag NPs with enzymes and proteins induces the production of ROS, which inhibits bacterial growth and colony formation, mainly through their reaction with DNA. Furthermore, PLGA was found to reduce the metal NPs’ aggregation and the Ag toxicity, which potentiates their antibacterial action. Ag-PLGA nanocomposites were shown to be suitable as a drug delivery system with high antibacterial activity, treating biofilm infections [19].

In a different scenario, PLGA/INP nanocomposites were shown to reduce ROS and counteract oxidative stress. In the work of Mehta et al. [167] PLGA MPs were used to encapsulate cerium oxide NPs (nanoceria—CNPs) together with SOD and catalase enzymes. Both the organic and the inorganic part of the system act synergistically to help prevent oxidative stress by scavenging ROS. In vitro studies have shown that the PLGA–nanoceria–SOD composites were efficiently delivered to macrophages and clearly displayed protection against environmental oxidative stress. These composites can potentially be applied to several diseases, such as neurodegenerative disorders and cardiovascular diseases, where oxidative stress plays a key role in the progression of the disease.

### 4.2. Diagnostic

Although theranosis is the most sought-after application involving imaging for PLGA-based materials, they have also been explored for purely diagnostic purposes, with one of the most common applications being MRI. This is made possible through the combination of PLGA with INPs with a magnetic character, which can either result in enhanced *T*_1_ or *T*_2_ MRI signals.

A good example was presented by Bennewitz et al. [146], which used PLGA-encapsulated MnO nanocrystals for cellular MR imaging. The longitudinal relaxivity, *r_1_* (a measure of the goodness of a contrast agent for *T*_1_-weighted MR imaging), of these composites was poor when the particles were intact. However, after incubation in acidic media the *r*_1_ increased (35-fold) due to the reduction and dissolution of MnO to free Mn^2+^. In the extracellular space the NPs do not release Mn^2+^ and consequently do not display a significative *T*_1_ contrast. On the other hand, when the composites were internalized by cells MR images showed enhanced *T*_1_ contrast due to the evolution of free Mn^2+^, allowing for the tracking of endocytosis [146]. In a different study by Nkansah et al. [123] PLGA micro- and nanoparticles were used for MRI-based cell tracking, this time through the encapsulation of magnetite. The magnetic PLGA composites’ degradation is faster in the first two weeks, displaying a quick decline in the *r_2_** relaxivity and then a gradual decline during the following 50 days. The composites were nontoxic, and mesenchymal stem cells labeled with magnetic PLGA composites presented a significant loss of *T_2_** signal as opposed to Feridex^®^ (a commercial MRI contrast agent) and blank PLGA-labeled cells. This study showed the utility of the presented composites as contrast agents for MRI-based cell tracking, being great candidates for clinical translation [123]. A similar study by Tang et al. [122] also proved the labeling efficiency of Fe_3_O_4_-encapsulating PLGA NPs.

In a different study Lu et al. [121] utilized PLGA MPs containing iron oxide for dual-modal PA/MRI tracking of tendon stem cells (TSCs) (Figure 5a). To enhance the cell internalization of the particles they were coated with poly-L-lysine, leading to a positive surface charge. It was demonstrated that the higher the Fe concentration the higher the PA and MRI signal achieved (Figure 5b). However, high concentrations of iron (>200 µg/mL) significantly reduced cell viability. Thus, an optimal concentration to maintain good PA and MRI signals for cell tracking without significant cytotoxicity was determined to be around 100 µg Fe/mL. It was observed that TSCs’ position could be tracked by monitoring the signal intensity at the anatomical sites where they accumulate. The main limitation of this study was that the labeling of TSCs was not detected in vivo. Nevertheless, the produced magnetic PLGA MPs demonstrated great potential to be used as a dual-mode imaging contrast agents.

The PLGA/INP combination for diagnosis is an ongrowing field that can be helpful for the study of various diseases where cell degeneration, mutation and growth is a determinant factor in disease evolution (e.g., cancer, neurodegenerative diseases). Extensive preclinical studies are already being developed in this field [185]; hence, it is of great importance to continue to explore it and eventually achieve the clinical translation of viable PLGA/INP formulations.

### 4.3. Theranostics

The effective prevention and treatment of diseases require the development of new strategies, with the aim of diagnosing and treating diseases at an early stage. PLGA/INPs composites that are able to act as multifunctional agents to achieve these ends have been extensively studied.

Lee et al. [127] performed an interesting conjugation of PLGA with complex molecules (mPEG and Ce6) and magnetite (core) to achieve a theranostic PLGA/INP composite. Ce6 and magnetite were the main functional parts of the system, with Ce6 working as a fluorescent and PDT agent able to generate ROS and magnetite working as an MRI contrast agent. It was observed that PLGA/INP composites with Ce6 molecules produced more ROS than free Ce6 when illuminated with a laser, resulting in a significant inhibition of the growth of tumors, it being the case that the tumor volume of mice treated with PLGA/INP composites was between ~1.5–3 times smaller than the ones treated with free Ce6. Even at low-dose administrations, PLGA composites with Ce6 were more efficient to cause tumor regression than free Ce6 administered at high doses. Consistent with this finding, PLGA/INPs showed a strong fluorescence signal at the tumor site, while free Ce6 had a very weak signal, proving the ability of multifunctional PLGA/INPs to accumulate in a tumor site. The encapsulation of Fe_3_O_4_ made the polymeric particles suitable for MRI imaging. In vivo studies showed an enhanced *T*_2_* contrast in the tumor when compared with Feridex^®^.

Once again, magnetite was the INP of choice in a study by Ye et al. [79], where PLGA vesicles containing SPIONs, Mn:ZnS QDs and a chemotherapeutic drug were tailored to be used as cancer theranostic agents. PLGA-SPION-Mn:ZnS vesicles exhibited a high *r*_2_* relaxivity and prominently enhanced the *T*_2_* signal in MR imaging. The signal decreased with the increase in the concentration of PLGA/INPs vesicles (and consequently of iron), showing that these hybrid composites can be used as negative contrast agents in MRI. Fluorescence imaging studies with macrophages facilitated by the presence of QDs showed that the composites had high uptake efficiency. The PLGA/INP vesicles could also be used as a DDS for the delivery of the lipophilic drug busulfan. The EE% of these drug was 89 ± 2%, and about 70–80% of the drug was released after the first 5 h of dialysis. Studies on the degradation of PLGA vesicles have suggested that 32% of PLGA degrades into LA and GA in 5 weeks. This relatively slow degradation behavior of PLGA vesicles combined with the high entrapment efficiency of busulfan makes this system ideal for sustained drug release, although further studies are needed to optimize the system release properties to match the degradation rate of PLGA.

In a different approach, Hao et al. [78] designed PEI-functionalized PLGA NPs combined with gold nanoshells, a targeting ligand (angiopep-2) and DTX, also to be used for cancer theranostics. The remote activation stimulus of an 808 nm laser on the Au nanoshell results in a heat-induced release of the cargo. The percentage of released drug increased by 17% compared with the release from nonirradiated composites. The PLGA/INP composites promoted efficient tumor inhibition with the help of angiogep-2 in addition to the enhanced permeability and retention effect. The nanocomposites also exhibited the potential for X-ray imaging applications. CT signal intensity increased with an increase in the concentration of composites. Overall, the core–shell DTX-loaded PLGA@Au nanocomposites represent a promising drug delivery system for tumor-targeted chemophotothermal therapy and X-ray imaging.

Song et al. [96] also proposed the conjugation of PLGA with gold in the form of AuNRs, where the gold (functionalized with PEG) was embedded in a PLGA shell. The AuNRs@PEG/PLGA nanocomposites showed good performance in PA image-guided photothermal therapy due to their optical properties, increased photothermal conversion efficiency and enhanced light scattering as well as PA signal (Figure 6a). The 60 nm composites showed an absorbance peak around 800 nm, which made them suitable for irradiation with an 808 nm laser. The photothermal conversion efficiency of the composites was up to twofold higher than that of AuNRs alone, due to the strong plasmonic coupling of the nanorods with the PLGA shell. When irradiated with the 808 nm laser the composites generated a PA signal and a temperature increase higher than that of free AuNRs. This temperature increase is essential for the photothermal destruction of the tumor, and the strong PA signal allows for the monitoring of this process. PA and PET (after labeling with Cu) imaging confirmed that after an IV injection the nanocomposites accumulated in the tumor region. Additionally, laser irradiation leads to the disassembly of the nanocomposites, where the PLGA shell degrades over time through the hydrolysis of PLGA ester bonds, resulting in AuNRs coated only with PEG. At day 10 postinjection the composites were already cleared from the body, which is essential for future clinical translations. These AuNR/PLGA nanocomposites were able to completely ablate tumors without reoccurrence in vivo (Figure 6b), presenting themselves as promising agents to be translated to the clinic in the near future for image-guided cancer therapy.

In the work by Topete et al. [97] a core–shell PLGA–gold nanocomposite was loaded with DOX and functionalized with HSA, noncovalently conjugated with indocyanine green (ICG) and covalently conjugated with folic acid (DOXO-loaded BGNSH-HSA-ICG-FA). The produced composites function as a drug release system of a chemotherapeutic drug through PTT. The gold nanoshell allows for the absorption of NIR (near-infrared) light, resulting in an increase in the localized heat that promotes efficient drug release, with irradiated samples releasing approximately 80% of the drug while nonirradiated samples only release approximately 20%. HSA grants stealth, while folic acid provides specific targeting. ICG is a NIR probe that can be used for PDT, where the absorption of NIR light results in the production of singlet oxygen that kills cancer cells. In addition, NIR probes can be used for in vivo imaging. Altogether, these nanoconjugates function as imaging agents and deliver a chemotherapeutic drug, heat and singlet oxygen directly to tumors. The in vitro treatment of cancer cells with the nanocomposites resulted in increased tumor cell death in the presence of NIR irradiation when compared to the respective controls (Figure 6c). This enhanced killing of tumor cells was the result of the synergistic effect of PDT/PTT. Furthermore, in vivo and in vitro studies showed the potential application of the nanoplatform as an imaging agent.

In another study, Wang et al. [175] also used AuNRs, but this time they were co-loaded with DTX in PLGA NPs coated with MnO_2_ nanofilms. The presence of Au and MnO_2_ in the PLGA/AuNR/DTX@MnO_2_ composites made them suitable for dual-mode imaging, with the Mn being able to work as a *T*_1_ contrast agent in MRI and the gold working as a positive X-ray CT imaging contrast agent (Figure 6d,e, respectively). Signal generation in both CT and MRI was indeed confirmed to be PLGA/INP-concentration-dependent. In addition, the MR signal derived from the MnO_2_ was enhanced in the presence of higher levels of glutathione (GSH), indicative of a responsive behavior that will specifically increase the contrast in the tumor (due to altered redox homeostasis). In terms of therapy, this study showed that the release rates from the PLGA/INPs were higher in acidic environments and with higher levels of GSH (from 39.7% to 78.1%). As with the responsive MR signal, this is due to the surface degradation of MnO_2_ and is a useful feature to add specificity to chemotherapeutic treatments. The presence of AuNRs in the particles also made them suitable for radiofrequency (RF) hyperthermia, which helps in the effectiveness of drug release. The combination of chemotherapy with the heat generated by the Au during RF hyperthermia allows for a reduction in the hypoxic area in the center of the tumor, since hypoxic cells are more sensitive to hyperthermia. The role of AuNRs in RF hyperthermia was proven in vitro, where it was demonstrated that cell viability significantly decreases when RF hyperthermia was applied in combination with the PLGA/AuNR/DTX@MnO_2_ (from approximately 60% to 20%). The same effect was observed in vivo. This indicates that AuNRs play a key role in this treatment and can in fact produce heat when irradiated with an RF pulse. Furthermore, AuNR-induced hyperthermia in combination with the effect of DTX responsive release effectively induced cellular apoptosis.

In summary, the combination of PLGA with INPs is a promising area, particularly in oncology theranostics, but it is of great importance to continue to optimize and develop new smart theranostic concepts and agents that have the potential to be translated to the clinic.

### 4.4. Tissue Engineering

The main application of PLGA in tissue engineering is through the development of PLGA scaffolds that are capable of inducing bone tissue regeneration to correct orthopedic defects, treat bone neoplasia and pseudoarthrosis or to stimulate recovery after surgery [11,186,187]. PLGA implants are growing rapidly as an alternative to traditional implants in many orthopedic applications. Since PLGA boasts a tunable degradation rate, it is possible to avoid further surgery to remove the implants, as in current clinical practice.

Mikael et al. [169] fabricated PLGA-MP-based scaffolds functionalized with inorganic MWCNT that render mechanically stronger scaffolds while maintaining good cellular compatibility for bone tissue engineering and regenerative engineering. The mechanical strength and compressive modulus of the scaffolds increased significantly with the addition of 1 and 3 wt% MWCNTs, since the nanotubes reinforced the junctions between microparticles. Furthermore, scaffolds made with a higher percentage of PLA (85:15 ratio) were mechanically stronger and degraded at a slower rate. PLGA/MWCNT composite scaffolds demonstrated their ability to act as substrates for HAp crystal growth, showing a higher deposition rate than that of pure PLGA scaffolds. In vivo studies did not show systemic or neurological toxicity. Furthermore, all implants maintained their structural integrity until week four, when they started to degrade. By weeks eight–twelve biological tissue started invading the implants. However, at this time point an inflammatory response was observed, which should be carefully studied before the translation of these composites scaffolds can move on.

Sheikh et al. [77] fabricated hybrid PLGA–silk fibroin scaffolds embedded with HAp NPs, which bring a beneficial synergistic effect from each of their components. PLGA–silk–HAp scaffolds present an improved swelling and water uptake capacity, which indicates that they can retain biological fluids, a beneficial feature, upon implantation. The inclusion of silk in the scaffolds presents several advantages: (i) it increases the hydrophilic character of PLGA, hence increasing the affinity towards biological fluids, allowing the cells to permeate deeper into the scaffold upon implantation; (ii) it improves the mechanical stability of the scaffold by penetrating PLGA micropores. Moreover, the incorporation of HAp NPs gives a higher stress-bearing capacity to the scaffold, improving its mechanical properties. In vitro studies have shown that the scaffolds had great cell infiltration, favorable for the intended application. In vivo studies have demonstrated that these scaffolds were able to complete intramembranous ossification at the site of a cavity due to the bone-inducing agent HAp. Following a very similar strategy, Selvaraju et al. [172] also prepared a hybrid composite scaffold with collagen (instead of silk) and PLGA embedded with HAp NPs. This study showed that HAp gives rigidity and stiffness to the nanocomposite. Besides enhancing the mechanical properties of the PLGA polymer, HAp also increases the thermal and conformational stability of collagen. The composite was nontoxic to healthy cells, being ideal to act as a scaffold. The degradation rate of the scaffold was studied for a period of 45 days and showed a minimum weight loss of 2%, which remained constant over time. Overall, both of these PLGA/INP scaffolds are great candidates for use in bone tissue engineering and clinical translation.

In a recent study, Pelaseyed et al. [170] created PLGA scaffolds (prepared as a foam) embedded with a TiO_2_ nanopowder. For applications in bone tissue regeneration the scaffolds should be highly porous (exhibiting micro- and macroporosity) to support cell seeding, adhesion and ingrowth proliferation. The incorporation of TiO_2_ into a PLGA matrix can improve these features, since PLGA itself is restricted by poor osteoconductivity. The prepared composite foam presented highly interconnected porous structures, which can be tuned by controlling PLGA concentration, freezing rate and cooling temperature. TiO_2_ NPs led to a decrease in pore size in the scaffolds. Additionally, the uniformly dispersed TiO_2_ particles in PLGA and the interaction between these two components improved the mechanical properties of the scaffolds. TiO_2_ in low concentrations (5 and 10 wt%) contributed to the decrease in the degradation rate of the polymer and the presence of this ceramic nanomaterial enhanced the bioactivity of PLGA. The prepared scaffolds were suitable for cell attachment, proliferation and nutrient transfer, and presented good mechanical properties, ideal again for applications in bone regeneration.

With a different goal in mind, Haider et al. [155] synthesized an antibacterial PLGA nanofiber scaffold composite incorporating CuO NPs. The fabricated hybrid PLGA/CuO nanocomposite fibbers were first studied as antibacterial materials through disc diffusion and optical density methods. In both methods the scaffolds produced zones of inhibition against *E. coli* and *S. aureus.* The hybrid scaffolds showed efficient inhibition of bacterial growth through the penetration of CuO or Cu^2+^ ions across the cell membrane as well as the generation of ROS; the adhesion of CuO NPs to proteins present in the bacterial cell wall caused bacterial death. The studies also proved that Cu^2+^ ions were released from the scaffolds in a sustained manner, enabling the antibacterial activity for longer periods of time. In vitro studies have shown that fibroblast cells had a good adherence, spreading and proliferating in the hybrid composite scaffolds, proving their good cytocompatibility and nontoxic nature. The results demonstrated that the PLGA/CuO nanofiber scaffolds have great potential to be used as internal and/or external wound dressing material.

Singh et al. [80] used PLGA MPs to produce hybrid PLGA scaffolds containing nanoceria, an INP that mimics the activity of SOD and catalase enzymes. In vitro studies have shown that the release of nanoceria from PLGA is slower than that of most drugs, since it relies on polymer degradation. The released INPs displayed a higher SOD activity in acidic media due to a higher presence of Ce ions. The produced MPs and scaffolds were biocompatible and the nanoceria particles were released in a slow and controlled fashion, being ideal as antioxidant treatments.

## 5. Conclusions

Currently, PLGA-based materials designed for biomedical applications are a subject of great interest, both at the research and industrial levels. The present review provides a compilation of several hybrid inorganic nanoparticle/PLGA-based composites in the nano-, micro- and macrorange and showcases the versatility of this family of materials. These hybrid materials have proved their superiority in terms of physicochemical features, biocompatibility, morphology and multifunctionality. Most of the discussed hybrid materials present a simple and straightforward synthesis, offering a great advantage over other kinds of materials since the synthesis can be easily adapted to fit a plethora of applications. The inorganic materials to be combined with PLGA often determine their final purposes, as do the organic therapeutic agents (proteins, DNA and drugs) that can be loaded onto the PLGA platform. Multifunctional PLGA-based composite materials allow for the combination of different applications that result in enhanced diagnostic and therapeutic outcomes. PLGA materials and nanocomposites represent a step forward in the biomedicine field and show great promise for further improvements in the theranostic, therapeutic, diagnostic and tissue engineering areas.

## Figures and Tables

**Figure 1 ijms-23-02034-f001:**
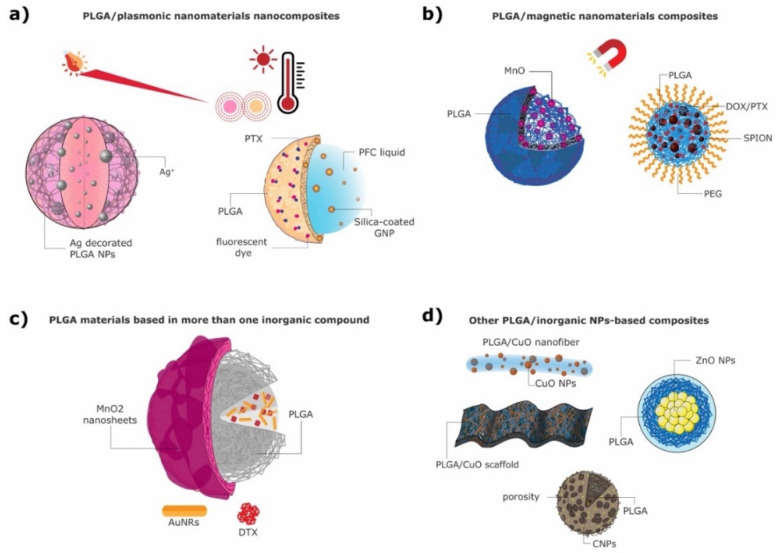
Different types of PLGA/INP nanocomposites: (**a**) plasmonic; (**b**) magnetic; (**c**) multifunctional (more than one inorganic compound); and (**d**) other nanocomposites.

**Figure 2 ijms-23-02034-f002:**
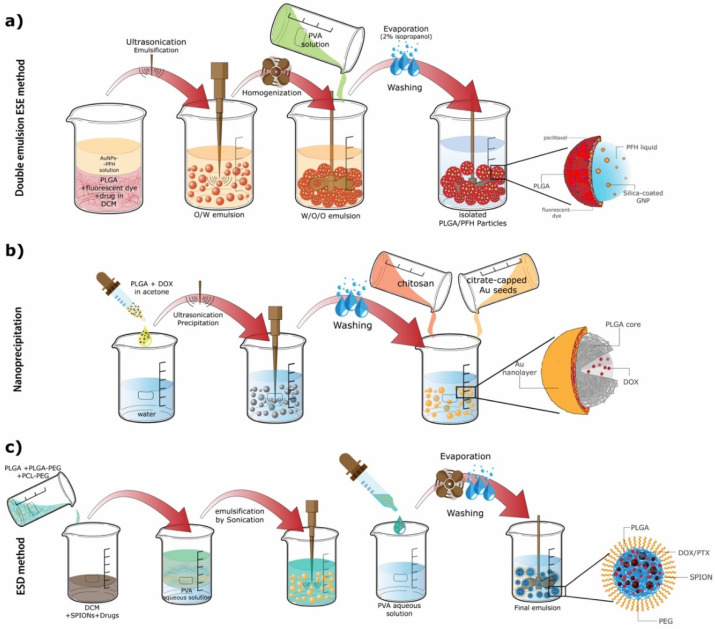
Illustration of (**a**) the double emulsion ESE method (**b**) the nanoprecipitation method and (**c**) the ESD method.

**Figure 3 ijms-23-02034-f003:**
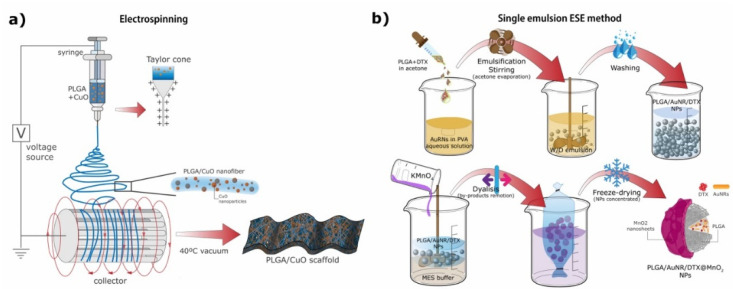
Illustration of (**a**) the electrospinning process and (**b**) the single emulsion ESE method as well as the further coating of the particles.

**Figure 4 ijms-23-02034-f004:**
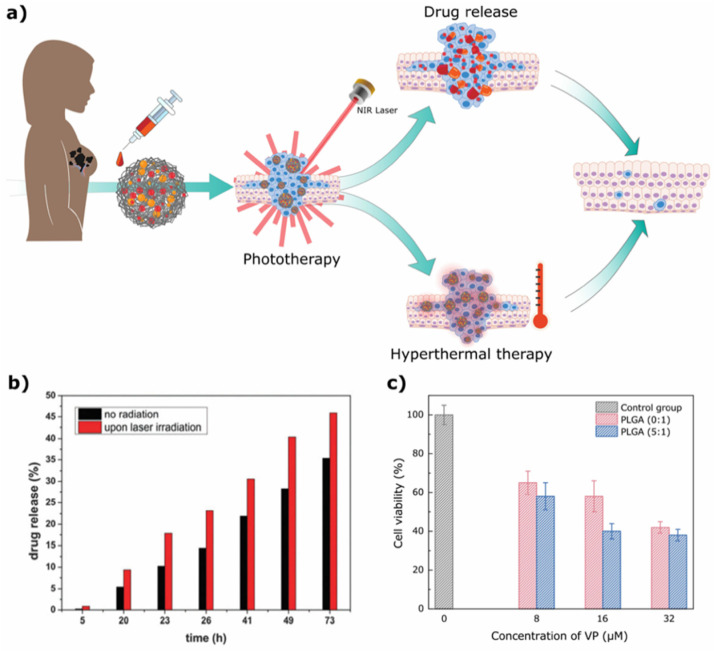
(**a**) Schematic representation of the applications of plasmonic PLGA NPs by Fazio et al. [95]. (**b**) Cumulative release from plasmonic PLGA NPs with and without laser irradiation. Reproduced with permission from ref. [95]. Copyright (2015), Royal Society of Chemistry. (**c**) Cytotoxicity of PLGA@AuNPs-VP by Deng et al. in PANC-1 cells after 5 min of 405 nm laser illumination. The molar ratios of Au and VP molecules in PLGA samples were 0:1 and 5:1, respectively. Adapted from ref. [94]. Copyright (2016), Royal Society of Chemistry.

**Figure 5 ijms-23-02034-f005:**
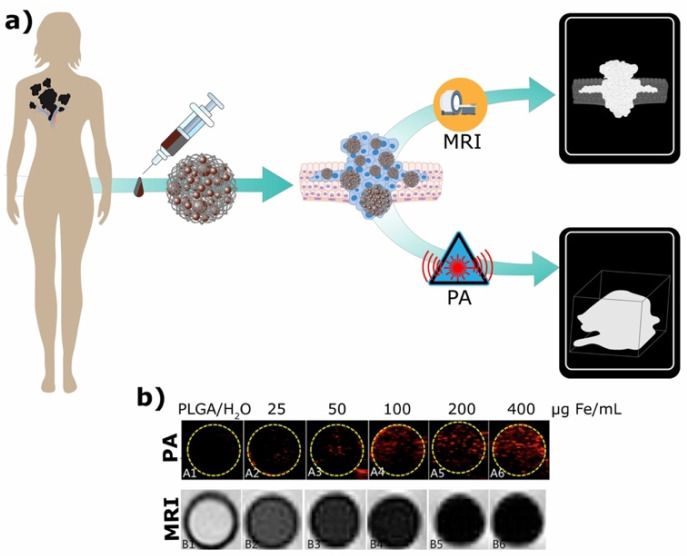
(**a**) Schematic representation of the applications of magnetic PLGA MPs by Lu et al. [121]. (**b**) Dual-modal PA/MRI imaging ability of magnetic PLGA MPs. Reproduced with permission from ref. [121]. Copyright (2018), Public Library of Science.

**Figure 6 ijms-23-02034-f006:**
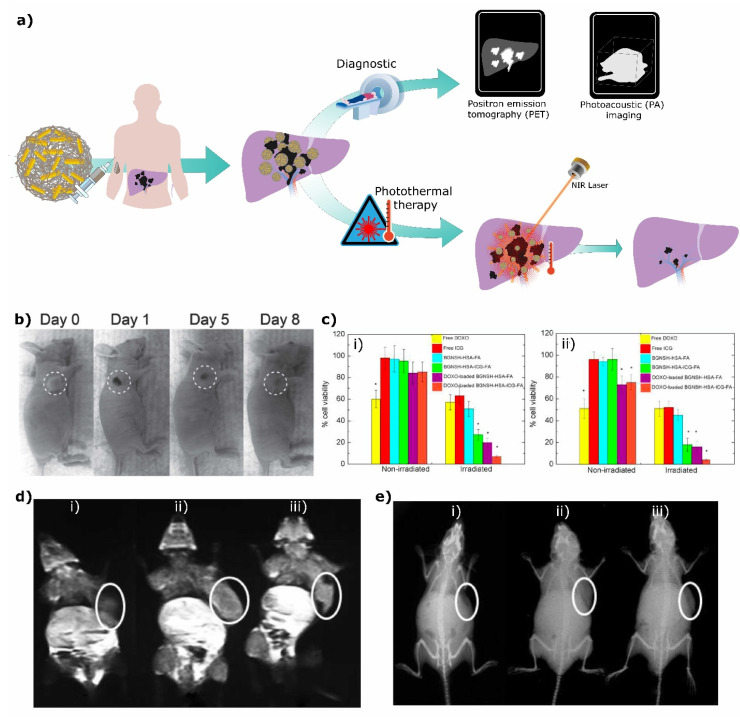
(**a**) Schematic representation of the applications of the AuNR vesicles from Song et al. [96]. (**b**) Photographs of the tumor-bearing mice at days 0, 1, 5 and 8 after being treated with the AuNR vesicles. Reproduced with permission from [95]. Copyright (2015), Wiley VCH. C. (**c**) Cell viability of (i) HeLa and (ii) MDA-MB-231cells after 24 h of incubation in the absence and presence of NIR light irradiation. Reproduced with permission from [97]. Copyright (2015), Wiley VCH; dual-mode imaging in vivo. (**d**) *T*_1_-weighted MR images and (**e**) X-ray CT images of (i) control, (ii) 4 h and (iii) 8 h after treatment. Reproduced with permission from [175]. Copyright (2017), Dovepress. * *p* < 0.05.

**Table 1 ijms-23-02034-t001:** PLGA-based nano-, micro- and macromaterials approved through the years for different biomedical applications. MP = microparticle; NP = nanoparticle; and VIP = vasoactive intestinal peptide.

Approval Year	Brand	Form/Active Principle	Route of Administration	Synthesis	Application
1989	Zoladex^®^	Implant/goserelin acetate	Subcutaneous	NA	Prostate carcinoma
1995	Lupron^®^	MP/leuprolide acetate	Intramuscular	W/O emulsion	Central precocious puberty/endometriosis
1997	Sandostatin^®^ LAR	MP/octreotide	Subcutaneous	Emulsification solvent evaporation method	Severe diarrhea associated with metastic tumors or VIP-secreting tumors
2002	Eligard^®^	Nanogel/leuprolide acetate	Subcutaneous	NA	Advanced prostate cancer
2002	Suprecur^®^	MP/buserelin acetate	Intramuscular	Spray-drying	Endometriosis
2003	Consta^®^	MP/risperidone	Intramuscular	Emulsification solvent evaporation method	Schizophrenia and bipolar disorder
2009	Ozurdex^®^	Implant/dexamethasone	Subcutaneous	NA	Macular edema
2014	Signifor^®^ LAR	MP/pasireotide pamoate	Intramuscular	NA	Cushing’s disease, acromegaly
2017	Zilretta^®^	MP/triamcinolone acetonide	Intra-articular	NA	Osteoarthritis
2017	Sublocade^®^	NP/buprenorphine	Subcutaneous	NA	Moderate to severe opioid addiction

## Data Availability

Not applicable.

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
