# Peer review of "PLGA-Based Composites for Various Biomedical Applications"

_ijms, 2022, doi:10.3390/ijms23042034_

Round 1

Reviewer 1 Report

please see attached pdf-file!

Author Response

The manuscript under review entitled “PLGA-based composites for various biomedical applications” by C.V. Rocha et al. is a review on the multitude of different PLGA-based nano-, micro- and macro-particles for the use in mainly biomedicine. The authors gathered information available in the literature about all the different aspects of the production procedures as well as the applications of these particles.

The authors start the review with a comprehensive introduction about the PLGA-based composites and restrict themselves to the combination of PLGA with inorganic nanoparticles, whereas they are also aware of organic combinations. The review is divided in three subtopics: Firstly, the properties, characterization and the different methods – emulsification-solvent evaporation, salting out, emulsification-solvent diffusion, nanoprecipitation, spray-drying and electrospinning – for the production of PLGA are described. Secondly, the different types of PLGA composite materials and their synthesis are given. In this part, already between the different applications – optical absorption related methods, photoacoustic methods, magnetic applications for imaging or e.g. hyperthermal theranostic and drug delivery – is distinguished. The third part is dedicated to the applications in biomedicine: therapy, diagnostic, theranostics and tissue engineering. Here, the authors show the great importance of optimization, functionalization and further development of the composite particles particularly in the field of oncology theranostics which is today more and more significant.

This manuscript collects all important aspects out of the multitude of papers available in the literature which is underlined by the high number of cited references.

We thank the reviewer for his/her thorough read of the manuscript and his/her positive comments on our work.

The manuscript should be accepted for publication after addressing the following minor notes:

Due to the amount of abbreviations used in the manuscript it would be very helpful for the reader to have a list of acronyms and abbreviations.

We thank this suggestion from the reviewer, which we agree that can be very helpful in review works like this one. We have now compiled an abbreviation list at the end of the manuscript.

There are some other minor issues which should be checked for the revision of the manuscript.

Line by line comments:

Line 42: typing error: which instead of wich

This typo has been corrected

Line 100: typing error? Do you mean tetrahydrofuran ?

This typo has been corrected

Line 144: typing error: … well established ….

This typo has been corrected

Line 242: typing error: I think you mean fibers. Fibber has a different meaning.

This typo has been corrected

Line 282: typing error: gathered instead of gattered

This typo has been corrected

Line 334: Please check the reference: Ref 103 is Esmaeili et al., not Hao et al. , because

In this case ref 103 is correct as Hao et al use Esmaeili’s synthesis in their preparation. Hao’s reference is then given a little bit later by the end of the paragraph.

Line 348: Missing reference number for Song et al.

The reference has been included

Line 383-384: should it be: …synthesized and functionalized …. ?

This typo has been corrected

Line 387: typing error? synthesized instead of synthetized ?

This typo has been corrected

Line 396: ROS – please explain the abbreviation ROS at first occurrence

We have now explained ROS

Line 417: missing reference number for Sun et al.

The reference has been included

Line 433 and line 521: it might be useful for reader (non-MRI expert) to explain T2 contrast and T1 contrast or to give at least a reference, e.g. https://mrimaster.com/characterise%20physics.html

A brief explanation of relaxation times has now been introduced in line 433.

Line 647: DL – please explain the abbreviation at first occurrence

We agree with the reviewer that abbreviations should be explained when you first use them. However in the mentioned case, DL is not properly an abbreviation (it is, it comes from the Latin dexter and laevus, right and left), it describes the chirality of the compound and it is always used like that. It is part of the polymer chemical nomenclature.

Line 766-768: inserted figure: Figure numbering is wrong: it must be Fig. 4. Figure and figure caption just copied from another paper?

Once again we thank the reviewer for his/her thorough review work. In this case the error must come from the editing process. We have now corrected it.

Line 794: you have explained r1, but no explanation for r2. Is it similar to the r1 explanation? The reviewer is correct, the explanation is the same.

Line 830: typing error: It was observed …

This typo has been corrected

Line 945: check reference: Ref. 186 is Karlsson et al.. Mikael et al. is missing.

The reference has been included

Reviewer 2 Report

There are no references in the manuscript. Authors are forgot to include the reference in the manuscript. Without that it is difficult to access the up to date account of this review. Hence I request the authors to provide the necessary references and upload revised file. 

Author Response

There are no references in the manuscript. Authors are forgot to include the reference in the manuscript. Without that it is difficult to access the up to date account of this review. Hence I request the authors to provide the necessary references and upload revised file.

We apologize to the reviewer for this crucial issue. We actually do not know what the problem with the references is. Reviewer 1 was able to see all the listed references without problem and when we downloaded the manuscript back to address reviewers comments, all references were there too. We have gone again through the references and we hope that this problem is now fixed.

Reviewer 3 Report

The review manuscript entitled "PLGA-based composites for various biomedical applications" by Rocha and co-workers summarize the literature data regarding PLGA and PLGA nanocomposites with different inorganic nanomaterials. The manuscript is logical and well organized, but I have some observations.

- First of all, the reference list is missing from the manuscript, and as this is a literature review, the evaluation of cited literature is very important to understand the information included in the review.

- The use of word “co-polymer” instead “copolymer”. I think that the second form is more common in the literature.

- On row 109, what you mean by “increasing in this way shelf stability”?

- On row 113 “biodegradable photoluminescent polyester”, which is the unit responsible for the photoluminescence properties?

- On row 830 “It was observed”

Author Response

The review manuscript entitled "PLGA-based composites for various biomedical applications" by Rocha and co-workers summarize the literature data regarding PLGA and PLGA nanocomposites with different inorganic nanomaterials. The manuscript is logical and well organized, but I have some observations.

We thank the reviewer for his/her positive comments on our work. We have tried to address all his/her comments one by one below.

- First of all, the reference list is missing from the manuscript, and as this is a literature review, the evaluation of cited literature is very important to understand the information included in the review.

We deeply regret this problem with the references. As mentioned above in the responses to reviewer 2, we do not know what the problem with the references is. Reviewer 1 was able to see all the listed references without problem and when we downloaded the manuscript back to address reviewers comments, all references were there too. We have gone again through the references and we hope that this problem is now fixed.

- The use of word “co-polymer” instead “copolymer”. I think that the second form is more common in the literature.

We have now used the form copolymer throughout our manuscript.

- On row 109, what you mean by “increasing in this way shelf stability”?

The idea that the original paper wanted to transmit and we wanted to convey with this expression is that the formulation could be stored for longer periods of time without losing its stability (e.g. the particles do not aggregate and sediment).

- On row 113 “biodegradable photoluminescent polyester”, which is the unit responsible for the photoluminescence properties?

We agree with the reviewer that it is not easy to identify a priory the emissive part of the polymer. This is what the authors reported in their work:

BPLPs are degradable oligomers synthesized from biocompatible monomers, including citric acid, aliphatic diols, and various amino acids via a convenient and cost-effective polycondensation reaction. BPLPs present some advantages over the traditional fluorescent organic dyes and quantum dots due to their cytocompatibility, minimal chronic inflammatory responses, controlled degradability, and excellent fluorescence properties [16]. Here, l-cysteine was selected and introduced into the polyester structure that was made of biocompatible monomers of citric acid and aliphatic1,8-octanediol, since previously reported BPLP from this starting amino acid exhibited the highest quantum yield (62.3%) [16]. The fluorophore structure of BPLP was verified as a fused ring structure ((5-oxo-3,5-dihydro-thiazolopyridine-3,7-dicarboxylic acid, TPA).

- On row 830 “It was observed”

This typo has now been corrected.

Round 2

Reviewer 2 Report

The paper contains lot of information from synthesis, composite preparation and their role in various biomedical applications. The authors have piled up the different methods to synthesize PLGA particles, composites a few in depth and few in vague. It seems to be just copied lot of information (PLGA, composite synthesis, diagnostic and therapeutic approaches). Overall it is one of a kind review article.  Few comments are as follows: 1. Reference style should be followed based on the journal style. 2. Figure taken from the published work should properly acknowledge. just a reference is not enough. Authors should state: Reproduced with permission from Ref. [XX]. Copyright (year), publisher name 3. Page 18: second paragraph:  There is a lack of clear evidence in distinguishing the Ag-PLGA composite role in bio-film treatment; The spaces within the sentence is not uniform seems to be copied from other source. please check plagiarism.   4. page 25; first paragraph (line 1006)In vitro studies need to be addressed well to conclude the composite as a Internal wound dressing and similarly external wound healing need to be explained. Discussion is missing  throughout the manuscript.  5. page 25; second paragraph: Based on SOD activity alone can't conclude as an ideal antioxidant composite more works need to support the statement.
The mechanism involved in it wasn't dealt properly. Authors should check other published article and give strong correlation.

Author Response

REVIEWER 2:

The paper contains lot of information from synthesis, composite preparation and their role in various biomedical applications. The authors have piled up the different methods to synthesize PLGA particles, composites a few in depth and few in vague. It seems to be just copied lot of information (PLGA, composite synthesis, diagnostic and therapeutic approaches). Overall it is one of a kind review article.

Few comments are as follows:

  1. Reference style should be followed based on the journal style.

We have now reformatted all the references

  1. Figure taken from the published work should properly acknowledge. just a reference is not enough. Authors should state: Reproduced with permission from Ref. [XX]. Copyright (year), publisher name

We have corrected the way in which we acknowledged figures from other authors.

  1. Page 18: second paragraph:  There is a lack of clear evidence in distinguishing the Ag-PLGA composite role in bio-film treatment; The spaces within the sentence is not uniform seems to be copied from other source. please check plagiarism. 

We have now modified the manuscript to better explain the role of AgNPs and we have included further references on this subject. Regarding the spacing and plagiarism comment, we have tried to summarise this work and its findings in our own words and with our own critical view.

  1. page 25; first paragraph (line 1006)In vitro studies need to be addressed well to conclude the composite as a Internal wound dressing and similarly external wound healing need to be explained. Discussion is missing  throughout the manuscript. 

Our intention with this review is to give the reader a flavour of the potential applications and advantages of PLGA-inorganic nanoparticles composites. The large number of works presented/discussed, which in our opinion are all relevant to this field, do not allow us to discuss in full detail all of the works. For further and more detailed information on a particular piece of work, references are given for the reader to find the original article and dive deeper into the subject of their interest.

  1. page 25; second paragraph: Based on SOD activity alone can't conclude as an ideal antioxidant composite more works need to support the statement.
    The mechanism involved in it wasn't dealt properly. Authors should check other published article and give strong correlation.

In the same direction as previous reply, SOD and antioxidant activity is not the focus of this review. We wanted to present the wide range of applications of PLGA composites in the biomedical area. For particular applications, we encourage the reader to search the provided references to learn more about each particular sub-field of research.

Reviewer 3 Report

The manuscript can be accepted in the present form.

Author Response

REVIEWER 3:

The manuscript can be accepted in the present form.

We thank the reviewer for his/her positive view of our manuscript